# Changes in Functionality of *Tenebrio molitor* Larvae Fermented by *Cordyceps militaris* Mycelia

**DOI:** 10.3390/foods11162477

**Published:** 2022-08-17

**Authors:** Neul-I Ha, Seul-Ki Mun, Seung-Bin Im, Ho-Yeol Jang, Hee-Gyeong Jeong, Kyung-Yun Kang, Kyung-Wuk Park, Kyoung-Sun Seo, Seung-Eon Ban, Kyung-Je Kim, Sung-Tae Yee

**Affiliations:** 1Department of Pharmacy, Sunchon National University, Jungang-Ro, Suncheon 549-742, Korea; 2Jangheung Research Institute for Mushroom Industry, Jangheung 59338, Korea; 3Suncheon Research Center for Bio Health Care, Jungang-Ro, Suncheon 57922, Korea

**Keywords:** edible insects, *Tenebrio molitor* larvae, functionality change, fermentation, *Cordyceps militaris* mycelia

## Abstract

The Food and Agriculture Organization (FAO) has been estimating the potential of insects as human food since 2010, and for this reason, *Tenebrio molitor* larvae, also called mealworms, have been explored as an alternative protein source for various foods. In this study, in order to increase nutrient contents and improve preference as an alternative protein source, we fermented the *T. molitor* larvae by *Cordyceps militaris* mycelia. *T. molitor* larvae were prepared at optimal conditions for fermentation and fermented with *C. militaris* mycelia, and we analyzed *T. molitor* larvae change in functionality with proximate composition, β-glucan, cordycepin, adenosine, and free amino acids content. *T. molitor* larvae fermented by *C. militaris* mycelia showed higher total protein, total fiber, and β-glucan content than the unfermented larvae. In addition, the highest cordycepin content (13.75 mg/g) was observed in shaded dried *T. molitor* larvae fermented by *C. militaris* mycelia. Additionally, the isolated cordycepin from fermented *T. molitor* larvae showed similar cytotoxicity as standard cordycepin when treated with PC-9 cells. Therefore, we report that the optimized methods of *T. molitor* larvae fermented by *C. militaris* mycelia increase total protein, total fiber, β-glucan and produce the amount of cordycepin content, which can be contributed to healthy food and increase *T. molitor* larvae utilization.

## 1. Introduction

As the world’s population steadily increases, the International Feed Industry Federation (IFIF) predicted that it will reach 10 billion by 2050 [1]. This means that it is necessary to increase food production to meet consumer demands. Mass breeding of livestock has a negative impact on the environment by producing a large amount of greenhouse gases and ammonia and spending considerable water, energy, and land [2]. A previous study reported that insects are significantly decreased the production of am-monia and greenhouse gases (CO_2_, N_2_O, and CH_4_) and also reported that less land-dependent than livestock [3,4]. In addition, insects are estimated to be able to supply food to approximately two billion people. For these reasons, edible insects have received worldwide attention as potential sources of alternative proteins [5,6].

Among the edible insects, *Tenebrio molitor* larvae, also called mealworms, are commercially reared in many countries as food for animals or even for human consumption. For example, *T. molitor* larvae are the most common insect produced in China, and the amount is more than 1000 tons per year [7,8]. In a previous study, *T. molitor* larvae were reported to have approximately 50% crude protein content [9] and contain essential amino acids, which are recommended for adult consumption by the World Health Organization (WHO) [7,10]. For these reasons, producers who cultivate *T. molitor* larvae and their production amounts have steadily increased. However, *T. molitor* larvae are usually produced in terms of living status and dried status, whereas consumers do not prefer them because of their disgusting appearance [11]. Therefore, it is necessary to develop a method that leads to an increase in *T. molitor* larvae consumption and utilization.

*Cordyceps militaris*, called Dong-Chong-Xia-Cao, which includes means of winter worm transforming into summer grass, is one of the entomogenous fungi that form a fruiting body mainly on pupae or larvae [12,13]. It has been traditionally used as an herbal medicine in Korea and China to enhance longevity and vitality [14] and contains many types of biologically active compounds such as cordycepin, adenosine, cordycepic acid, sterols, nucleosides, and polysaccharides [15]. Among them, cordycepin (3′-deoxyadenosine), a derivative of adenosine, was the first compound to be isolated from *C. militaris* [16]. Cordycepin has been reported to have antitumor, antiangiogenic, antimetastatic, and antiproliferative effects, and it induces apoptosis in cancer cells [17].

Biofortification is a very important method to improve the shortage of food and provide nutrients for those who rarely have access to foods [18]. Among those, fermentation is a very economical and useful method of improving nutrient ingredients by producing active ingredients [19,20]. Fermentation is a process that is subjected to the action of microorganisms or enzymes, and it can cause biochemical changes in food [21]. These changes may result in sensory qualities and improved nutritional value [22]. A previous study that fermented the *T. molitor* larvae performed lactic acid bacteria and soy sauce fermentation, but there is not enough study fermented with *C. millitaris* myceila using their entomogenous character. On account of this reason, in this study, *T. molitor* larvae fermented by *C. militaris* mycelia were performed to increase their utilization and progressing appearance. In addition, we analyzed changes in functionality to confirm the potential of nutritional components as food.

## 2. Materials and Methods

### 2.1. Samples

*T. molitor* larvae used in this study were purchased from Myeong Pum, Inc. (Jang-seong, Korea). *T. molitor* larvae were reared for 9 weeks and did not feed for two days before harvest to remove feces. Harvested *T. molitor* larvae were separated from wheat and excretions using steel mesh. *T. molitor* larvae were separated into two test groups to compare the nutrient composition and functional ingredient and to find optimal fermentation conditions. One test group of *T. molitor* larvae was spread in a steel tray and dried under shade for 20 h at room temperature (shade dried, SD). The other test group of *T. molitor* larvae was also spread in a steel tray and dried in an oven for 20 h at 80 °C and boiled for 30 min (30 min boil after hot air dried, 30BHAD).

### 2.2. Chemicals and Standards

Analytical-standard cordycepin and adenosine were purchased from Sigma-Aldrich (St. Louis, MO, USA). Standards of amino acids, OPA was purchased from Agilent (Agilent Technologies, Santa Clara, CA, USA). Certified HPLC-grade solvents methanol and acetonitrile were purchased from J. T. Baker Chemicals (Center Valley, Coopersburg, PA, USA). All other reagents used for analysis were analytical grade. Potato dextrose agar (PDA) and potato dextrose broth (PDB) were purchased from BD DIFCO (Detroit, MI, USA).

### 2.3. Fermentation

The *C. militaris* strain (No. 40226) was obtained from the Korean Agricultural Culture Collection (KACC, Suwon, Korea) and used in this study. To prepare the inoculum, *C. militaris* mycelia in the PDA plates were punched using a cork borer (diameter 5 mm), and five discs were transplanted into 500 mL PDB and incubated for 10 days at 25 °C with 150 rpm in a shaking incubator. The 500 mL liquid media were transferred to 1 L PDB, incubated for 15 days at 25 °C, and homogenized for transplantation. *T. molitor* larvae of two test groups (SD, 30BHAD) were filled with each polypropylene bottle and sterilized in an autoclave for 30 min at 121 °C. The polypropylene bottle was cooled to room temperature on a clean bench and inoculated with an equal ratio of inoculum (*v*/*w*, 1:2). The polypropylene bottles containing *T. molitor* larvae and *C. militaris* mycelia were harvested after incubation for 60 days at 25 °C and freeze-dried for use in the analysis.

### 2.4. Proximate Composition Determination

The contents of proximate compositions were measured according to the official methods of the Association of Official Analytical Chemists (AOAC, Rockville, Maryland, 2005). The moisture content was determined by oven-drying methods at 105 °C for 12 h. Crude protein (N × 6.25) in the samples was determined using the Kjeldahl method. Crude fat was extracted using a Soxhlet apparatus. To measure the ash content, samples were burned at 600 °C for 3 h. The crude fiber was extracted as digesting with dilute H_2_SO_4_ and KOH solutions.

### 2.5. β-Glucan Determination

The β-glucan content was measured using a Megazyme yeast and mushroom kit (Megazyme, Bray, Ireland), and β-glucan content was calculated as the total glucan content minus the α-glucan content.

### 2.6. Analysis of Free Amino Acids

To determine free amino acids, samples were extracted for 3 h in 60 °C water bath with distilled water. The extraction was filtered using a syringe filter of 0.45 μm (Advantec Dismicr, Tokyo, Japan) and incubated at 4 °C after adding sulfosalicylic acid. After 4 h, water extract was separated by centrifugation at 10,000× *g* for 10 min and used for analysis. The free amino acids were quantified by high-performance liquid chromatography (HPLC; 1200 Series HPLC system, Agilent Technologies, Santa Clara, CA, USA) after derivatization with ο-phthaldialdehyde. The column size was 2.1 × 150 mm, with a 4 µm particle size (Agilent Technologies, Santa Clara, CA, USA) and a 340 nm detection wavelength at 40 °C. Mobile phase A was 10 mM sodium phosphate and 10 mM sodium tetraborate (*v*/*v*, 1:1), pH 8.2, and mobile phase B was acetonitrile, methanol, and distilled water (*v*/*v*/*v*, 45:45:10) with a flow rate 0.35 mL/min.

### 2.7. Analysis of Cordycepin and Adenosine

To determine cordycepin and adenosine, samples were extracted using a sonicator for 90 min in distilled water. The water extract was separated by centrifugation at 2264× *g* for 15 min. The supernatant was filtered using a syringe filter of 0.45 μm (Advantec Dismicr, Tokyo, Japan). HPLC analysis was performed on a 1200 Series HPLC system (Agilent Technologies, Santa Clara, CA, USA) with an Agilent ZORBAX Eclipse XDB-C_18_ column (Agilent Technologies, Santa Clara, CA, USA). The parameters used were as follows: flow rate of 0.8 mL/min, injection volume of 10 μL, wavelength of 260 nm, and column temperature of 30 °C.

### 2.8. Separation and Purification Cordycepin

Separation and purification were performed to determine the cell viability effect of cordycepin isolated from SD *T. molitor* larvae fermented by *C. militaris* mycelia. Semi-HPLC was used for separation. The composition ratio of the mobile phase was LC-water and methanol (*v*/*v*, 70:30) to obtain a sub-fraction by loading the extract of SD *T. molitor* larvae fermented by *C. militaris* mycelia into the cartridge column. Among them, the main fraction that included the cordycepin peak was collected and concentrated under freeze drying and gained 46.5 mg of dried cordycepin peak fraction. HPLC-MS/ELSD was used for purification. The dried peak fraction and cordycepin standard were dissolved in methanol and analyzed using an Agilent Poroshell 120 SB—C_18_. Detection was performed under the condition that the composition ratio of the mobile phase was water: Acetonitrile (*v*/*v*, 70:30), and the flow rate was 0.4 mL/min. The injection volume was 10 μL, and the detection wavelength was 260 nm.

### 2.9. Cell Culture and Viability Assay

PC-9 cells (human lung adenocarcinoma cell line) were obtained from the European Collection of Authenticated Cell Cultures (ECACC). The cells were cultured in RPMI-1640 medium containing 10% fetal bovine serum, 1% penicillin/streptomycin, and 0.1% 2-mercaptoethanol. Cultures were maintained at 37 °C and 5% CO_2_, and the media were changed every three days. Cells were seeded in a 96-well plate (6 × 10^4^ cells/well) and subcultured for 24 h, and 10, 30, 100, and 300 µM isolated cordycepin and standard cordycepin were treated to each well. The cells were incubated in 5% CO_2_ at 37 °C for 48 h. Subsequently, 100 µL of the solution was removed from each well, and 10 µL of CCK-8 reagent was added. After 1 h, the absorbance was measured at 450 nm using a microplate reader (Versamax, Molecular Devices, San Jose, CA, USA).

### 2.10. Statistical analysis

Data are presented as mean ± SD of three replicates. Statistical differences between groups were compared using the Student’s *t*-test. Probability values less than 0.05, were considered significant (*p* values * < 0.05, ** < 0.01, and *** < 0.001). One-way ANOVA and Tukey’s post hoc test were performed using Graphpad 7.0 (Graphpad Software, San Dieogo, CA, USA).

## 3. Results

### 3.1. Proximate Composition and β-Glucan Contents

Proximate compositions were analyzed to determine the usefulness of *T. molitor* larvae fermented by *C. militaris* mycelia as a functional food source. Table 1 shows the proximate composition of *T. molitor* larvae fermented with or without *C. militaris* mycelia. The proximate composition of SD *T. molitor* larvae and fermented by *C. militaris* mycelia contained moisture (3.5% and 9.15%), crude protein (44.01% and 59.34%), crude fat (32.02% and 18.54%), crude ash (3.21% and 3.84%), and crude fiber (5.54% and 9.48%, respectively). In addition, 30BHAD *T. molitor* larvae fermented by *C. militaris* mycelia contained moisture (2.25% and 8.87%), crude protein (47.49% and 64.24%), crude fat (36.9% and 32.02%), crude ash (2.63% and 2.21%), and crude fiber (12.97% and 15.62%, respectively). By fermentation of *T. molitor* larvae with *C. militaris* mycelia, they substantially increased the crude protein and crude fiber content and reduced the crude fat content. In a previous study, it was reported that bioactive compounds such as ergosterol, the precursor of vitamin D_2_, are integrally linked with mycelial growth [23]. The protein and fat contents of several mushrooms grown in various media have been described previously [24]. Microbial proteins can be derived from a variety of microorganisms, both unicellular and multicellular, including bacteria, yeast, and fungi [25]. In this study, *T. molitor* larvae fermented by *C. militaris* mycelia produced more crude protein than unfermented *T. molitor* larvae. We suggest that these results are affected by microbial proteins and can be more attractive as alternative protein sources than unfermentation.

β-Glucan contents of *T. molitor* larvae fermented by *C. militaris* mycelia are shown in Table 1. First, different content of β-glucan was shown between SD *T. molitor* larvea (0.34%) and fermented SD *T. molitor* larvea (4.77%). The β-glucan content of 30BHAD *T. molitor* larvae was 0.61%, and when 30BHAD *T. molitor* larvae fermented by *C. militaris* mycelia, it contained 8.78%, which the highest β-glucan content in the samples. The β-glucan content of *T. molitor* larvae fermented by *C. militaris* mycelia was substantially higher than those of the non-fermented *T. molitor* larvae. It is well known that mushrooms contain many bioactive compounds that have diverse beneficial impacts on human health [26]. In particular, β-glucan is the main polysaccharide normally present in the cell walls of fungi and yeast. It has been reported that β-glucan affects immune system activation, as well as antimicrobial, antioxidant, and antitumor effects [27,28]. According to these results, we expect the use of *T. molitor* larvae as functional food.

### 3.2. Free Amino Acids Content

The free amino acids content of *T. molitor* larvae before and after fermentation are shown in Table 2. In all of the samples, a total of 15 amino acids, which include eight essential and seven non-essential amino acids, were detected. Total free amino acids content of SD *T. molitor* larvae was 2587.73 mg/100 g and after fermentation contained 1626.05 mg/100 g. Moreover, total free amino acids content of 30BHAD *T. molitor* larvae was 1080.14 mg/100 g and after fermentation contained 3018.25 mg/100 g. In case of SD *T. molitor* larvae, total free amino acids content was 0.62-fold decreased after fermentation, but 30BHAD *T. molitor* larvae was 2.79-fold significantly increased. Total essential amino acids content of SD *T. molitor* larvae was 1138.64 mg/100 g but decreased after fermentation to 607.33 mg/100 g (0.53-fold). On the other hand, the total essential amino acids content of 30BHAD *T. molitor* larvae was 490.88 mg/100 g and after fermentation increased to 1262.5 mg/100 g (2.57-fold). *C. militaris* is entaentomogenous fungi that form a fruiting body mainly on pupae or larvae and use insects as a nutrient source for grown [12,13]. For these reasons, it seems that the free amino acids content of SD *T. molitor* larvae fermented by *C. militaris* mycelia was decreased [29]. On the other hand, the total free amino acids in 30BHAD *T. molitor* larvae was increased after fermentation. These free amino acids content differences require additional experiments. Other previous studies showed that when *Pleurotus eryngii* and *Bacillus subtilis* were fermented in *Gryllus bimaculatus*, total free amino acids increased due to protease [29,30]. Additionally, aspartic acid content of 30BHAD *T. molitor* larvae was 28.72 mg/100 g and after fermentation increased to 149.61 mg/100 g. The glutamic acid content of 30BHAD *T. molitor* larvae was 143.68 mg/100 g and after fermentation increased to 588.89 mg/100 g. These free amino acids are known for being highly responsible for the taste properties and development of flavor-forming reactions [31]. Aspartic acid and glutamic acid are related to umami [32], umami content of 30BHAD *T. molitor* larvae was significantly increased when fermented by *C. militaris* mycelia.

### 3.3. Cordycepin and Adenosine Contents

Table 3 shows the cordycepin and adenosine contents of *T. molitor* larvae fermented with or without *C. militaris* mycelia. Cordycepin and adenosine contents of unfermented *T. molitor* larvae were not detected, and when SD *T. molitor* larvae fermented by *C. militaris* mycelia, it contained 13.75 mg/g of cordycepin and 0.85 mg/g of adenosine. Additionally, 30BHAD *T. molitor* larvae fermented by *C. militaris* mycelia contained 9.19 mg/g of cordycepin and 1.66 mg/g of adenosine. The cordycepin and adenosine contents of *T. molitor* larvae were produced according to fermentation.

Additionally, we purchased commercial *C. militaris* to compare cordycepin and adenosine contents with SD and 30BHAD *T. molitor* larvae fermented by *C. militaris* mycelia. Commercial *C. militaris* were separated from the fruiting body and substrate for analysis. Table 3 also shows the cordycepin and adenosine contents of commercial *C. militaris*. The contents of cordycepin in commercial *C. militaris* were found to be highest in order to CCMP substrate (6.45 mg/g), CCMP fruiting body (1.06 mg/g), CCMBR2 fruiting body (1.16 mg/g). Furthermore, the adenosine contents in commercial *C. militaris* were found to be highest in order to CCMP fruiting body (4.1 mg/g), CCMBR2 fruiting body (3.24 mg/g), CCMBR1 fruiting body (2.77 mg/g). In the case of CCMBR1, 2 cordycepin and adenosine contents of fruiting bodies were higher than those of the substrate. Additionally, the adenosine content of the CCMP fruiting body was higher than the substrate, but cordycepin content was higher in the fruiting body. Among the different substrates of *C. militaris*, the SD *T. molitor* larvae had the highest cordycepin content (13.75 mg/g). Cordycepin, which was first isolated from *C. militaris*, has been reported to have antitumor, antiangiogenic, antimetastatic, and antiproliferative effects [16,17]. Previous studies have reported cordycepin content in *C. militaris* fruiting bodies of brown rice were 5.62 mg/g [33], 1.2–3.5 mg/g [34], and in a substrate of brown rice was 2.27–3.97 mg/g [35]. A previous study reported that the differences between cordycepin contents are due to differences in the *C. militaris* strain [35]. This study showed that *T. molitor* larvae produce amount of cordycepin when fermented by *C. militaris* mycelia. According to a previous study, cordycepin is well known for strong antitumor effects. Therefore, we expect an additional bio-active effect of *T. molitor* larvae fermented by *C. militaris* mycelia.

### 3.4. Cell Viability Assay

We used the human lung cancer cell line PC-9 to investigate the cell viability of the isolated cordycepin produced in *T. molitor* larvae. Standard cordycepin and isolated cordycepin were administered at 10, 30, 100, and 300 μM for 48 h. The CCK-8 assay showed a dose-dependent decrease in the growth of PC-9 cells. The toxicity of standard and isolated cordycepin at 100 μM was 42.4% and 61%, respectively; at 300 μM, it was 31% and 42.7%, respectively (Figure 1). Cordycepin is well known for cytotoxicity in various cancers, and PC-9 cells are more sensitive to lung cancer [36]. In this study, we confirmed the cordycepin isolated from SD *T. molitor* larvae fermented by *C. militaris* mycelia reduces PC-9 cell viability as well as standard cordycepin.

## 4. Conclusions

This study was performed to analyze the functional change of *T. molitor* larvae according to fermentation by *C. militaris* mycelia. The functional changes of *T. molitor* larvae were investigated using two drying methods (SD and 30BHAD) and fermentation with C. militaris mycelia. The results showed not only did *T. molitor* larvae increase crude protein, crude fiber, β-glucan, and free amino acids (only 30BHAD *T. molitor*), but it also reduced crude fat when fermented with *C. militaris* mycelia. Moreover, *T. molitor* produced cordycepin and adenosine after fermentation, and also SD *T. molitor* larvae fermented by *C. militaris* mycelia produced the highest contents of cordycepin compared with commercial *C. militaris*. Then, we isolated cordycepin from *T. molitor* larvae, which has the highest content, and analyzed its cell viability activity; it inhibited the growth of PC-9 cancer cells. In this study, we only performed cytotoxicity effects of isolated cordycepin from fermented *T. molitor* larvae, but if additional anti-cancer effects are confirmed, it can be possible to develop functional foods using fermented *T. molitor* larvae. These results suggest that fermentation with *C. militaris* mycelia can increase the functionality of *T. molitor* larvae. In addition, fermentation using *C. militaris* mycelia to *T. molitor* larvae is considered to will be useful for the development of high-protein and eco-friendly alternative foods.

## Figures and Tables

**Figure 1 foods-11-02477-f001:**
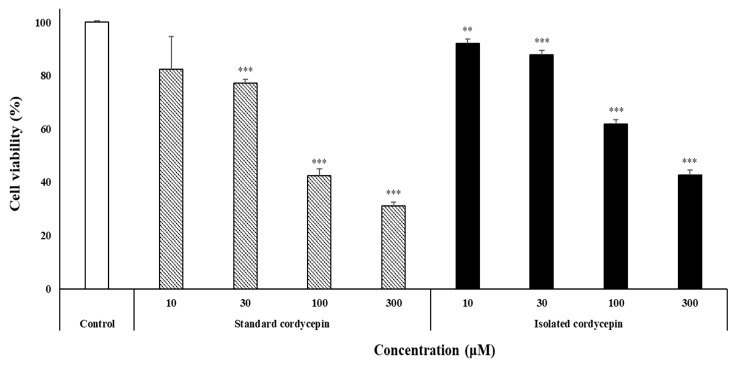
Effect of standard cordycepin and isolated cordycepin on PC-9 human lung cancer cell viability. Cells were seeded at a density of 6 × 10^3^ cells/well in a 96-well plate and then treated with different concentrations of standard cordycepin and isolated cordycepin (10, 30, 100, and 300 µM) for 48 h. The culture supernatant was removed, and cell counting kit-8 was added. All data are expressed as mean ± SD of three independent experiments. ** *p* < 0.01, *** *p* < 0.001 compared to the control.

**Table 1 foods-11-02477-t001:** Proximate composition and β-glucan content of shade dried (SD), and 30 min boil after hot air dried (30BHAD) *Tenebrio molitor* larvae before and after fermentation.

Variables (%)	SD		30BHAD	
	Unfermented	Fermented	Unfermented	Fermented
Moisture	3.5 ± 0.24 ^a1^	9.15 ± 0.79 ^b^	2.25 ± 0.09 ^c^	8.87 ± 0.15 ^b^
Crude protein	44.01 ± 0.27 ^a^	59.34 ± 0.12 ^b^	47.49 ± 0.19 ^c^	64.24 ± 0.52 ^d^
Crude fat	32.02 ± 0.21 ^a^	18.54 ± 1.24 ^b^	36.9 ± 0.58 ^c^	32.02 ± 0.21 ^d^
Crude ash	3.21 ± 0.04 ^a^	3.84 ± 0.19 ^a^	2.63 ± 0.42 ^ab^	2.21 ± 0.15 ^b^
Crude fiber	5.54 ± 0.24 ^a^	9.48 ± 0.2 ^b^	12.97 ± 1.44 ^c^	15.62 ± 0.19 ^d^
β-Glucan	0.34 ± 0.17 ^a^	4.77 ± 0.26 ^b^	0.61 ± 0.29 ^a^	8.78 ± 0.33 ^c^

^1^ All values are means ± SD (N = 3). ^a–d^ Groups that do not share a common letter indicate a significant difference (*p* < 0.05).

**Table 2 foods-11-02477-t002:** Free amino acid contents of shade dried (SD), and 30 min boil after hot air dried (30BHAD) *Tenebrio molitor* larvae before and after fermentation.

Variables (mg/100 g)	SD		30BHAD	
	Unfermented	Fermented	Unfermented	Fermented
Aspartic acid	91.22 ± 11.49 ^a^^1^	84.8 ± 14.16 ^a^	28.72 ± 4.33 ^b^	149.61 ± 7.44 ^c^
Glutamic acid	224.22 ± 37.16 ^a^	292.31 ± 44.24 ^a^	143.68 ± 30.04 ^a^	588.89 ± 67.85 ^b^
Serine	81.47 ± 8.26 ^a^	59.53 ± 7.67 ^ab^	40.66 ± 7.24 ^b^	88.61 ± 11.54 ^a^
Histidine	165.14 ± 16.02 ^a^	81.7 ± 10.16 ^b^	52.18 ± 11.6 ^b^	128.87 ± 19.16 ^a^
Glycine	48.02 ± 3.56 ^a^	25.04 ± 2.94 ^b^	21.45 ± 2.95 ^b^	36.38 ± 5.56 ^c^
Threonine	77.35 ± 8.61 ^a^	52.28 ± 5.87 ^b^	38.44 ± 6.85 ^b^	112.77 ± 9.37 ^c^
Arginine	456.76 ± 42.2 ^a^	385.09 ± 35.44 ^a^	172.39 ± 32.27 ^b^	602.94 ± 85.19 ^c^
Alanine	239.22 ± 31.98 ^a^	56.68 ± 13.94 ^b^	122.08 ± 25.07 ^b^	98.04 ± 38 ^b^
Tyrosine	308.19 ± 42.63 ^a^	115.27 ± 12.17 ^b^	60.27 ± 11.24 ^b^	191.28 ± 24.32 ^c^
Valine	189.16 ± 30.24 ^a^	113.28 ± 11.29 ^b^	91.54 ± 16.9 ^b^	221.11 ± 18.24 ^a^
Methionine	63.82 ± 5.39 ^a^	33.36 ± 5.16 ^b^	18.48 ± 2.09 ^c^	26.16 ± 1.9 ^bc^
Phenylalanine	120.66 ± 12.59 ^a^	49.37 ± 5.59 ^b^	51.32 ± 9.62 ^b^	102.14 ± 11.98 ^a^
Isoleucine	124.7 ± 16.84 ^a^	48.77 ± 5.07 ^b^	59.51 ± 10.6 ^b^	70.32 ± 4.85 ^b^
Leucine	143.62 ± 16.6 ^a^	57.47 ± 5.58 ^b^	91.01 ± 17.92 ^bc^	117.56 ± 16.43 ^ac^
Lysine	254.18 ± 70.69 ^a^	171.1 ± 4.31 ^a^	88.41 ± 33.91 ^a^	483.57 ± 114.86 ^b^
Total free AA	2587.73 ± 330.84 ^a^	1626.05 ± 160.85 ^b^	1080.14 ± 219.62 ^b^	3018.25 ± 419.4 ^a^
Total EAA	1138.64 ± 162.42 ^a^	607.33 ± 41.08 ^b^	490.88 ± 107.59 ^b^	1262.5 ± 188.49 ^a^

^1^ All values are means ± SD (N = 3). ^a–c^ Groups that do not share a common letter indicate a significant difference (*p* < 0.05).

**Table 3 foods-11-02477-t003:** Cordycepin and adenosine contents of various *Cordyceps militaris,* including *Tenebrio molitor* larvae before and after fermentation.

Samples		Variables (mg/g)	
		Cordycepin	Adenosine
SD	Unfermented	ND ^1^	ND
	Fermented	13.75 ± 0.03 ^a^^3^	0.85 ± 0.03 ^a^
30BHAD	Unfermented	ND	ND
	Fermented	9.19 ± 0.02 ^b^	1.66 ± 0.05 ^b^
CCMBR1 ^2^	Fruiting body	0.78 ± 0.02 ^c^	2.77 ± 0.03 ^c^
	Substrate	0.26 ± 0.01 ^d^	0.4 ± 0.01 ^d^
CCMBR2	Fruiting body	1.16 ± 0.02 ^e^	3.24 ± 0.03 ^e^
	Substrate	0.63 ± 0.01 ^f^	0.23 ± 0.01 ^f^
CCMP	Fruiting body	1.06 ± 0.04 ^g^	4.1 ± 0.09 ^g^
	Substrate	6.45 ± 0.02 ^h^	1.7 ± 0.12 ^b^

^1^ ND—not detected. ^2^ CCMBR1—commercial *C. militaris* grown on brown rice 1; CCMBR2—commercial *C. militaris* grown on brown rice 2; CCMP—commercial *C. militaris* grown on the pupae. CCMBR1,2 were purchased from distinct farmer market. ^3^ All values are means ± SD (N = 3). ^a–h^ Groups that do not share a common letter indicate a significant difference (*p* < 0.05).

## Data Availability

Data are contained within the article.

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
