# Peer review of "Changes in Functionality of *Tenebrio molitor* Larvae Fermented by *Cordyceps militaris* Mycelia"

_foods, 2022, doi:10.3390/foods11162477_

Round 1
Reviewer 1 Report
The manuscript number: foods-1870944 entitled “Changes in Functionality of Tenebrio molitor Larvae Fermented by Cordyceps militaris Mycelia” authored by Neul et al., is the comparative study of the bio-fortification of nontraditional food for human nutritional security. The author bio-fortificatified Tenebrio molitor larvae by using Cordyceps militaris mycelia. They reported that the Tenebrio molitor larvae fermented with Cordyceps militaris mycelium have higher total protein, fiber, and β-glucan content than the un-fermented larvae. In addition, they also reported higher cordycepin content in fermented larvae.
The reviewer went through the manuscript and found that the manuscript is presented as per journal guidelines, the contents of manuscript is in defined format, table and figures are presented appropriately. Followings are the query and suggestion for improving the quality of the manuscript
1. Abstract is more introductory, focus should be emphasized on results so may be modified.
2. Keywords are not catchy, add a few more like “edible insects, bio-fortification, value addition” etc.
3. Introduction is written nicely but the author should briefly add the importance of food bio-fortification. For this author can see “Jaiswal, D. K., Krishna, R., Chouhan, G. K., de Araujo Pereira, A. P., Ade, A. B., Prakash, S., ... & Verma, J. P. (2022). Bio-fortification of minerals in crops: current scenario and future prospects for sustainable agriculture and human health. Plant Growth Regulation, 1-18.”
4. Method and materials section written nicely.
5. Results and discussion are also nicely written and the interpretation of results in the discussion section is properly presented.
6. The author should rewrite the “conclusion” section for more soundness; focus should be on application of present study findings to set a future roadmap for food and nutritional security to the increasing global population.
The present study will open a new avenue of bio-fortification of nontraditional foods for food and nutritional security. As the global agricultural land and resources are shrinking day by day and population increasing continuously. Therefore, there is an urgent need for the popularization of nonconventional food for food and nutritional security. Therefore, the manuscript may be considered for publication in foods for publication after minor English corrections.
Author Response
Response to Reviewer 1 Comments
Point 1: Abstract is more introductory, focus should be emphasized on results so may be modified.
Response 1: As suggested by the reviewer, we modified abstract to focus on results with more details. (Lines: 15-19) ,(Lines: 24-26)
“In this study, in order to increase ----- larvae by Cordyceps militaris mycelia.”, “Also, the isolated cordycepin ----- when treated with PC-9 cells.”
Point 2: Keywords are not catchy, add a few more like “edible insects, bio-fortification, value addition” etc.
Response 2: As suggested by the reviewer, we added keyword. (Line: 30-31)
“Edible insects”
Point 3: Introduction is written nicely but the author should briefly add the importance of food bio-fortification. For this author can see “Jaiswal, D. K., Krishna, R., Chouhan, G. K., de Araujo Pereira, A. P., Ade, A. B., Prakash, S., ... & Verma, J. P. (2022). Bio-fortification of minerals in crops: current scenario and future prospects for sustainable agriculture and human health. Plant Growth Regulation, 1-18.”
Response 3: Thanks for the careful checking. We added contents of food bio-fortification with reference. (Lines: 64-67)
“Biofortification is very important ----- by producing active ingredients [19, 20].”
Point 4: Method and materials section written nicely.
Response 4: Thanks for your kindness. We really glad to hear that. Thank you.
Point 5: Results and discussion are also nicely written and the interpretation of results in the discussion section is properly presented.
Response 5: Thanks for your kindness. We really glad to hear that. Thank you.
Point 6: The author should rewrite the “conclusion” section for more soundness; focus should be on application of present study findings to set a future roadmap for food and nutritional security to the increasing global population.
Response 6: Suggested by the reviewer, we modified conclusion to highlight the sample as a alternative foods. (Lines: 304-307)
“These results suggest that fermentation ----- and eco-friendly alternative foods.”

Reviewer 2 Report
Dear Editor and Author,
The article entitled “Changes in Functionality of Tenebrio molitor Larvae Fermented by Cordyceps militaris Mycelia” is interesting as this paper discuss about the feasibility of insect (i.e., Tenebrio molitor Larvae) as new source of protein. Some analyses used sophisticated equipment. However, some issues must be solved by authors prior to the acceptance for publication in this journal:
1. There is a serious issue in the English editing. Please check carefully. Going through professional proofreader is highly recommended. For examples: line 18-20, line 36, etc. Error in using English makes the articles is not easy to follow.
2. Line 49-50: it will be much better if the authors show the disgusting appearance of Tenebrio molitor Larvae. Also, please the English or local name of Tenebrio molitor in the manuscript. It will be easier for readers to follow.
3. Introduction section: please explain why the Tenebrio molitor need to be fermented? What is the outcome of fermentation process based on the theory? Previous studies related to fermentation must be referenced to support the aims of this research. What is the novelty of this research?
4. Line 67: What is the reason of not feeding the larvae 2 days before harvesting?
5. Line 109: please convert rpm to g
6. Table 2 can be combined with Table 1
7. Table 4 and Table 5 can be combined. But please do pivoting of the tables (column to be row; and vice versa).
8. Table 1-5, where is the result of statistical analysis? Please show in the table.
9. Conclusion section: Please mention the limitation of this study, and then give some advises for further research.
Author Response
Response to Reviewer 2 Comments
Point 1: There is a serious issue in the English editing. Please check carefully. Going through professional proofreader is highly recommended. For examples: line 18-20, line 36, etc. Error in using English makes the articles is not easy to follow.
Response 1: As suggested by the reviewer, we modified mentioned sentence. And we also announce to you that manuscript was professional proofreads (site: https://www.editage.co.kr/, order number : TIVCP_1). (Lines: 19-21, 38-41)
‘’In this study, in order to increase ------and fermented with C. militaris mycelia.”,“A previous study reported ----- land-dependent than livestock [3, 4].”
Point 2: Line 49-50: it will be much better if the authors show the disgusting appearance of Tenebrio molitor Larvae. Also, please the English or local name of Tenebrio molitor in the manuscript. It will be easier for readers to follow.
Response 2: As suggested by the reviewer, we added english name of Tenebrio molitor larvae. And we also suggested GA for show the disgusting appearance of Tenebrio molitor Larvae. (Line: 16) (Line: 44)
“The Food and Agriculture Organization (FAO) --------- also called mealworm have been gaining as an alternative protein source for various foods.”, ” Among the edible insects, Tenebrio molitor larvae also called mealworm -------- human consumption.”,
Point 3: Introduction section: please explain why the Tenebrio molitor need to be fermented? What is the outcome of fermentation process based on the theory? Previous studies related to fermentation must be referenced to support the aims of this research. What is the novelty of this research?
Response 3: Thanks for the careful checking. We added contents of fermentation with the reference and added this study's purpose. (Lines: 65-75)
“Among those, fermentation is --------of nutritional components as food.”
Point 4: Line 67: What is the reason of not feeding the larvae 2 days before harvesting?
Response 4: Thanks for the careful checking. Because we have to remove the feces. And added in sentance. (Lines: 80)
“T. molitor larvae were reared for -------- to remove feces.”
Point 5: Line 109: please convert rpm to g
Response 5: We modified the unit of rpm to unit of g and also changed the manuscript. (Lines: 120-121)
“After 4 h, water ex-tract was------at 10,000 g for 10 min and use for analysis.”
Point 6: Table 2 can be combined with Table 1
Response 6: Thanks for the careful checking. As suggested by the reviewer we combined table 1 and 2. (Line: 203)
Point 7: 4 and Table 5 can be combined. But please do pivoting of the tables (column to be row; and vice versa).
Response 7: We combined table 4 and 5 and switched the column and row. (Line: 246)
Point 8: Table 1-5, where is the result of statistical analysis? Please show in the table.
Response 8: Thanks for the careful checking. We were performed statistical analysis of all tables by one-way ANOVA and Tukey’s post hoc test. And added values on tables. (Lines: 203, 234, 246)
Point 9: Conclusion section: Please mention the limitation of this study, and then give some advises for further research.
Response 9: Thanks for the careful checking. The sentence including limitation of this study and for the further research has been inserted. (Lines: 301-304)
“In this study, we only performed cytotoxicity effects ---------- foods using fermented T.molitor larvae.”
